# Targeted Therapy for EWS-FLI1 in Ewing Sarcoma

**DOI:** 10.3390/cancers15164035

**Published:** 2023-08-09

**Authors:** Helong Gong, Busheng Xue, Jinlong Ru, Guoqing Pei, Yan Li

**Affiliations:** 1Department of Orthopaedic Surgery, Shengjing Hospital, China Medical University, No. 36 Sanhao Street, Heping District, Shenyang 110004, China; hlgong@cmu.edu.cn; 2Department of Hematology, The First Affiliated Hospital of Xi’an Jiaotong University, Xi’an 710061, China; bushengxue@xjtufh.edu.cn; 3Institute of Virology, Helmholtz Centre Munich, German Research Centre for Environmental Health, 85764 Neuherberg, Germany; jinlong.ru@helmholtz-munich.de; 4Department of Orthopedics, Xijing Hospital, Air Force Medical University, Xi’an 710032, China; guoqpei@163.com

**Keywords:** ewing sarcoma, EWS-FLI1, protein complex, targeted therapy, immunotherapy

## Abstract

**Simple Summary:**

Ewing sarcoma (EwS) is a highly aggressive and metastatic cancer in children and adolescents. Canonical therapy mainly comprises the combination of intensive chemotherapy, radiation, and local surgery, which give rise to acute and chronic adverse effects. Drugs targeting EwS without side effects are in urgent demand. Genetically, EwS is characterized by chromosomal translocations with a low mutation burden. As a result, the chimeric protein EWS-ETS, mainly EWS-FLI1(85%), is critical for the malignancy of EwS. EWS-FLI1 directly binds to GGAA microsatellites in enhancers and promotors of the target genes and recruits multiple transcription factors or epigenetic regulators to reprogramme the epigenome. Direct targeting EWS-FLI1 is difficult due to the disordered structure, we mainly review the current knowledge of EWS-FLI1 property, the EWS-FLI1 protein complex, and the downstream pathways, we also summarize the targeted therapy of EwS by taking advantage of the EWS-FLI1 protein complex and the immunotherapy of the genes activated by EWS-FLI1.

**Abstract:**

Ewing sarcoma (EwS) is a rare and predominantly pediatric malignancy of bone and soft tissue in children and adolescents. Although international collaborations have greatly improved the prognosis of most EwS, the occurrence of macrometastases or relapse remains challenging. The prototypic oncogene EWS-FLI1 acts as an aberrant transcription factor that drives the cellular transformation of EwS. In addition to its involvement in RNA splicing and the DNA damage response, this chimeric protein directly binds to GGAA repeats, thereby modifying the transcriptional profile of EwS. Direct pharmacological targeting of EWS-FLI1 is difficult because of its intrinsically disordered structure. However, targeting the EWS-FLI1 protein complex or downstream pathways provides additional therapeutic options. This review describes the EWS-FLI1 protein partners and downstream pathways, as well as the related target therapies for the treatment of EwS.

## 1. Introduction

Ewing sarcoma (EwS) is a poorly differentiated malignancy that mainly arises in bone and soft tissue and is more prevalent among those in the second decade of life [1]. Extraosseous EwS is more common in adults [2]. The cellular origin of EwS remains controversial, although it is speculated that it arises from neuroectodermal cells or primitive mesenchymal stem cells (MSC) [3,4,5]. Despite considerable improvements in overall survival achieved using a multimodal approach, including intensive chemotherapy for localized disease [6], the prognosis of patients who develop metastatic disease or relapse remains dismal [7,8]. Approximately 20–25% of patients are diagnosed with early micro-dissemination [6,9].

Dissecting the pioneer molecular events provides a precision approach to the treatment of cancers including EwS. EwS is genetically characterized by chromosomal translocation, which generates the EWS-ETS (85% is EWS-FLI1) chimeric protein accompanied by a quiet genomic background [10,11]. EWS-ETS governs the transcriptome to drive malignant transformation and serves as the most reliable diagnostic marker of EwS, as well as providing a therapeutic target. Directly targeting EWS-ETS is difficult because of its intrinsically disordered structure [12,13]; therefore, understanding the EWS-ETS protein complex and the downstream molecular events involved in EwS is essential for the design of therapeutic strategies.

EwS is genetically distinct because of a balanced chromosomal translocation that joins the N-terminus of EWS (on chromosome 22) with a gene from the C-terminus of the E-twenty-six (ETS) family of transcription factors (FLI1, ERG, ETV1, ETV5, and FEV), predominantly FLI1 (85%, on chromosome 11) and ERG (10%, on chromosome 21) [14]. This has been well reviewed in the literature [1,15,16,17].

The EWS protein consists of an N-terminal transcription-activation domain (Exons 1 to 7) and C-terminal RNA-binding domains (Exons 8 to 17) [18]. The N-terminus contains a serine–tyrosine–glycine–glutamine (SYGQ) domain that has low-complexity and is intrinsically disordered. The C-terminus contains three RNA-binding arginine–glycine–glycine- (RGG-) motifs, the three RGG- motifs are separated by an RNA-recognition motif (RRM) and a zinc finger motif (ZF) [19]. EWS functions in transcription, RNA processing, transportation, alternative splicing, DNA repair, and homologous recombination [20,21,22,23,24]. Because of these multiple functions, EWS is essential for the central nervous system [25], meiosis, B-lymphocyte development, and hematopoietic stem-cell self-renewal [22]. The ETS domain of the ETS protein binds to DNA sequences with a central GGAA/T core motif [26], and the protein is thus involved in cell-cycle regulation and differentiation [27,28].

Chromosomal translocation results in the formation of a chimeric protein because the RNA-binding domain of EWS is replaced by a DNA-binding domain [14]. The formation of this chimeric protein is thought to be the initial event in the tumorigenesis and malignancy of EwS.

## 2. Involvement of EWS-FLI1 in Transcription, Epigenetic Reprogramming, and Alternative Splicing in EwS

### 2.1. EWS-FLI1 in Transcription and Epigenetic Reprogramming

EWS-FLI1 is an aberrant transcription factor that drives cellular transformation by rewiring the epigenome to induce a large number of oncogenes. The N-terminus of EWSR1 contains a prion-like domain, characterized by an intrinsically disordered structure and low complexity. This domain has phase transition properties and manipulates multiple proteins involved in epigenome reprogramming and epigenetic alterations [29,30,31,32,33,34,35,36]. In addition to the canonical ETS-binding sites, EWS-FLI1 binds to DNA sequences at the GGAA/T core motif [37,38,39] via a conserved ETS domain. It regulates multiple proteins through its prion-like domain to tumor-specific enhancers and promotors, recruiting acetyltransferases and establishing de novo enhancers by generating H3K27ac, thus opening the chromosomal architecture, which contributes to the activation of target genes [29,30,37,39]. The EWS-FLI1 protein complex includes RNA polymerase II [23,40], the core subunit hsRBP7 (human RNA polymerase II) [41,42], E2F3 [43,44], EWSR1 [45], CBP/p300 [46], WDR5, ASH2, MLL [30], and the BAF complex (mammalian SWI/SNF complex) [29,47]. The threshold of GGAA motifs optimal for maximal expression is 20–26 [48], which differs from that in wild-type FLI1. Super-enhancer-associated MEIS1 and RING1B also contribute to the chromatin reprogramming through co-localization with EWS-FLI1 at the active enhancers to drive the malignancy of EwS [49,50]. This specific coupling results in the activation of many genes (Figure 1), such as NKX2.2 [51], NROB1 [52,53,54], IGF1R [55], BCL11B [56], EZH2 [36], VRK1 [30], GLI1 [57], PTPL1 [58], PPPR1A [59], ERG2 [60], GSTM4 [61], PAX7 [62], CHM1 [63], REST [64], PHF19 [32], STEAP1 [65,66], SLFN11(Schlafen 11) [67], HDAC3 [68], TNC [69], APCDD1 [49], IL1RAP [70,71], MYC [72], and PRC1 (protein regulator of cytokinesis 1) [73].

Among the direct targets of EWS-FLI1, NKX2-2 mediates oncogenic transformation via transcriptional repression and is necessary and sufficient for the oncogenic phenotype of EwS [74]. Further work demonstrates that NKX2-2, KLF15, and TCF4 occupy similar super-enhancers and promoters, forming an inter-connected auto-regulatory loop and occupying 77.2% of promoters and 55.6% of enhancers shared with EWS-FLI1 [75], such as STEAP1 [76]; this kind of coordinate regulation drives the proliferation of EwS. NROB1 directly interacts with EWS-FLI1 to modulate multiple gene expressions and mediate the oncogenic phenotype of EwS [77]. SLFN11 is a putative DNA/RNA helicase that recruits to the stressed replication fork and irreversibly triggers replication block and cell death. Overexpression of SLFN11 is associated with resistance to topoisomerase I inhibitors and poly (ADP-ribose) polymerase (PARP) inhibitor combinations [67,78]. STEAP1 and IL1RAP are vital for the redox homeostasis of EwS [65,70]. APCDD1, PHF19, GSTM4, and PTPL1 are genes that are involved in the proliferation of EwS.

EWS-FLI1 is also involved in transcriptional repression of tumor suppressors such as IGFBP3 [79] and PHLDA1 [53] to drive oncogenic transformation [31,80]. The nucleosome remodeling and deacetylase (NuRD) complex is a typical ATP-dependent chromatin remodeling complex [81] that plays a critical role in transcription and determines differentiation and development [82]. EWS-FLI1 recruits the NuRD-LSD1 complex to repress LOX and TGFBR2 [80,83]. It also affects the transcriptional activation of AP-1 [33] and MRTFB [84] and binds to the promotor of FOXO1 to repress its expression, thereby increasing tumor growth [85]. EWS-FLI1 promotes the phosphorylation of cyclin-dependent kinase-2 and AKT to inhibit the activity of FOXO1, thus rewiring transcriptional repression [85]. EWS-FLI1 is also involved in the regulation of microRNAs (miRNAs) [86]. It downregulates miRNA-145 to initiate mesenchymal stem-cell reprogramming toward EwS stem cells [87] and represses miR-708, which induces the overexpression of EYA3 and contributes to the chemoresistance to etoposide and doxorubicin [88].

The histone methyltransferase EZH2 exhibits silencing activity via methylation of H3K27 [89]. In EwS, EWS-FLI1 upregulates EZH2 expression by interacting with the EZH2 promoter, thereby promoting tumor growth/metastasis and blocking endothelial/neuro-ectodermal differentiation [36].

MiR-34a inhibits the proliferation and increases the sensitivity of EwS to doxorubicin and vincristine and is a strong predictor of a favorable prognosis in EwS [90]. However, the exact mechanism underlying its downregulation remains elusive. Exportin 5 (XPO5), which mediates the nuclear export of pre-miRNAs and short hairpin RNAs [91,92,93], interacts with EWS-FLI1 based on mass spectrometry [94]. XPO5 is highly expressed in various cancers including EwS (Appendix A). Furthermore, the phosphorylation of XPO5 alters the nucleus and cytoplasm shift [95]. Investigating XPO5 and its relationship with EWS-FLI1 may offer new insights into the therapy of EwS. Post-translational modifications of EWS-FLI1 modulate its transcriptional activity. Phosphorylation and O-GlcNAcylation of the N-terminus of EWSR1 [96,97,98], as well as acetylation of the C-terminal FLI1 domain by PCAF (KAT2B, lysine acetyltransferase 2B), enhance the transcriptional activity of EWS-FLI1 [99]. However, PCAF expression is lower in EwS tissues, which is a common feature of cancer (Appendix A).

### 2.2. EWS-FLI1 in Alternative Splicing

Pre-mRNA splicing is critical for gene expression, and most protein-encoding transcripts are alternatively spliced to provide diverse functions [100,101]. The N-terminus of EWSR1 interacts with the hyperphosphorylated RNA polymerase II and recruits serine-arginine (SR) through its C-terminus. After chromosome translocation, the C-terminus of wild-type EWSR1 is replaced by FLI1, which hinders the recruitment of SR-splicing factors and interferes with mRNA splicing [23], thus demonstrating the negative property of this chimeric protein [102]. This leads to comprehensive alternative splicing of numerous genes. Meanwhile, EWS-FLI1 interacts with the splicing components (snRNP) U1C and SF1 to modulate pre-mRNA splicing [103]. It also recruits the BAF complex to drive the alternative splicing of ARID1A and the preferential splicing of ARID1A-L, which is necessary for tumor growth [104]. Work by Selvanathan [94] demonstrates that EWS-FLI1 is involved in the alternative splicing of CLK1, CASP3, PPFIBP1, and TERT, which potentially regulate the oncogenesis of EwS.

## 3. The Regulation of EWS-FLI1

Transcription and post-transcriptional modifications are involved in the regulation of expression and activity of EWS-FLI1. Although the transcriptional regulation of EWS-FLI1 remains elusive, the BRD4 inhibitor JQ1 suppresses this activity [32,34,105]. HDAC6 deacetylates specificity protein 1 (SP1), thereby inhibiting the recruitment of the SP1/P300 complex to the promoters of EWSR1 and EWS-FLI1 and downregulating EWS-FLI1 [106]. MiR-145 and let-7 repress EWS-FLI1 by targeting its mRNA [87,107,108,109] and inhibit the proliferation of EwS. The RNA-binding protein LIN28B interacts with EWS-FLI1 transcripts to maintain the stability and ensure the expression of EWS-FLI1 to enhance the tumorigenicity of the self-renewal of EwS [108]. At the post-transcriptional level, EWS-FLI1 degradation is proteasome dependent, and the protein has a half-life of 2–4 h [110]. This process can be protected by the action of ubiquitin-specific protease 19 (USP19) at the N-terminus [111] and accelerated by tripartite-motif-containing 8 (TRIM8) at K334 [112]; however, USP19 is expressed at low levels in EwS (Appendix A). Casein kinase 1 (CK1)-mediated phosphorylation of the VTSSS degron in the FLI1 domain activates speckle-type POZ protein (SPOP) activity, which degrades EWS-FLI1. In contrast, OTU-domain-containing protein 7A (OTUD7A) participates in the deubiquitination of the C-terminus and stabilizes EWS-FLI1 [113]. The inhibitor of chromosomal maintenance 1 (CRM1 and XPO1), KPT-330 [114], and IFN-γ [115] suppress expression of EWS-FLI1 at the protein level. FOXM1, a downstream factor of EWS-FLI1, upregulates its expression [116]. Cytosine arabinoside (ARA-C) downregulates EWS-FLI1 at the protein level and inhibits tumor growth [117]; however, it shows hematologic toxicity and minimal activity in patients [118].

STAG2 (stromal antigen 2) is a core subunit of the cohesion complex and is frequently mutated in multiple cancers [119] including EwS [10,120]. Mutation of STAG2 in EwS is associated with poor outcomes by improving metastasis [10]. Mechanically, in addition to the disruption of PRC2-mediated regulation of gene expression in EwS [121], the inactivation of STAG2 strongly altered CTCF-anchored loop extrusion and decreases promotor-enhancer interactions. As a result, the cis-mediated EWS-FLI1 activity at GGAA microsatellite neo-enhancers is downregulated and the cells are enhanced in their migration and invasion properties [121,122].

Unlike STAG2, there was no evidence showing mutations of the ETS transcription factor ETV6 in EwS [123]. ETV6 does not change the expression of EWS-FLI1 but co-occupy loci genome wide at the short consecutive GGAA repeats and constrains the transcriptional activity of EWS-FLI1 [124,125]. Upon inactivating ETV6, EWS-FLI1 overtakes and activates these cis-elements to promote mesenchymal differentiation by upregulating the expression of SOX11 [125].

## 4. EWS-FLI1 in the Metastasis and DNA Damage Repair of EwS

### 4.1. EWS-FLI1 in the Metastasis of EwS

Tumor metastasis is a major cause of death in cancer patients, and the survival of EwS patients with early metastasis is poor [126,127], especially those with bone metastasis alone or with multiple metastatic sites [9,128]. In addition to the tumor microenvironment, the differences in EWS-FLI1 expression levels determine the balance between proliferation and metastasis in EwS [129,130]. EWS-FLI1^high^ cells promote proliferation, whereas EWS-FLI1^low^ cells tend to shift to a cell-matrix adhesion phenotype, which gives rise to invasion/metastasis by altering the structure of the cytoskeleton [84,129,131,132] and eventually drives metastatic-related death. Many genes repressed by EWS-FLI1 are enriched in focal adhesion, including zyxin and α5-integrin [133]. EWS-FLI1 also suppresses the MRTFB/YAP/TAZ/TEAD regulatory pathway by inhibiting the Rho-F-actin signal, resulting in the inhibition of the invasive properties of EwS [84,131]. On the other hand, the canonical targets of EWS-FLI1 such as DKK2 [134], CAV1 [135,136], EZH2 [36], CHM1 [63], GPR64 [137], PPP1R1A [59], and TNC [69] are involved in the invasive properties of EwS.

Hypoxia is a common feature of cancers [138], and hypoxia-inducible factor-1a (HIF-1a) is highly expressed in EwS and contributes to the metastasis of EwS [139]. HIF-1a also activates the transcription of EWS-FLI1. However, cells induced by hypoxia are mainly enriched in an HIF-1a signature in EwS, indicating that the treatment of EwS should not only focus on EWS-FLI1, but also on the microenvironmental conditions such as hypoxia [139].

AMER2 is a direct target of EWS-FLI1 [108] (Figure 2A), and knockdown of EWS-FLI1 inhibits the expression of AMER2 based on GSE176190 (Figure 2B). AMER2 is highly expressed in EwS compared to other cancers based on the CCLE database (Appendix A) and normal tissues based on GSE17679 (Figure 2C). Activation of Wnt/β-catenin signaling is a common feature in multiple metastatic cancers [140,141], whereas AMER2 works as a negative regulator of Wnt/β-catenin signaling [142]. The knockdown of EWS-FLI1 might activate Wnt/β-catenin and drive the metastasis of EwS (Figure 2D), but the specific mechanisms are still to be explored.

### 4.2. EWS-FLI1 in DNA Damage Repair

In addition to its involvement in regulating and driving EwS, EWS-FLI1 causes DNA damage [24], leading to growth arrest and apoptosis [112,143,144,145]. EWS-FLI1 binds to the promotor of caspase-3; conditional expression of EWS-FLI1 in embryos increases caspase-3 expression and triggers early onset of apoptosis and acute lethality in mouse kidney [144]. The hypersensitivity to radiation [146] and DNA-damaging agents such as etoposide [24,147,148] provides further evidence that tailored therapies based on DNA-damaging agents [6] are needed, despite the fact that few mutations have been identified that participate in DNA damage repair [149].

The role of PARP in the DNA damage response has been studied extensively, and its inhibition is widely used in breast and ovarian cancer with BRCA1/BRCA2 mutations [150,151,152]; PARP inhibitors mainly target defective homologous recombination [153]. In EwS, EWS-FLI1 maintains the expression of PAPR1 and also forms a protein complex with PARP1, which promotes oncogene-dependent sensitivity to PARP1 inhibition [148]. EWS-FLI1 also inhibits EWSR1 in a dominant–negative manner [154,155]. EWSR1 interacts with PARylated PARP1 through RGG domains at DNA damage sites and suppresses DNA damage by preventing excessive PARP1 accumulation. However, in EwS, the presence of EWS-FLI1 leads to the hyper-PARylation and accumulation of PARP1 [156]. On the other hand, EWS-FLI1 promotes the aberrant transcription by reversing EWSR1-dependent inhibition of CDK9 activity, resulting in the accumulation of R-loops and blocking the release of BRCA1 from transcription complexes, finally leading to transcription stress. This mimics BRCA1 deficiency and impairs homologous recombination repair [24]. Paronetto et al. demonstrated that EWSR1 depletion results in the alternative splicing of genes involved in DNA repair and genotoxic stress signaling, including ABL1, CHEK2, and MAP4K2, which decreases cell viability and proliferation upon UV irradiation [157]. After chromosomal rearrangement, wild-type EWSR1 is replaced by EWS-FLI1, which might mimic the depletion of EWSR1. EWS-FLI1 dysregulates the expression and the function of PARP1 and is involved in the hypersensitivity of EwS to DNA-damaging agents.

EWSR1 binds to negatively charged poly-ADP ribose molecules via positively charged RGG repeats [158,159] and then recruits to the DNA double strand; however, after the formation of EWS-FLI1, the RGG repeats in the C terminus are replaced. Menon et al.’s work shows that EwS are distinct from BRCA1/2 mutant tumors and do not display the genomic scars of homologous recombination [160]. The aberrant recruitment of EWS-FLI1 to sites of DNA damage can disrupt physiologic DNA double-strand break repair. They also found that EWS-FLI1 suppressed the activation of DNA damage sensor ATM; meanwhile, a collateral dependence on the ATR signaling axis was induced by EWS-FLI1, and EWS-FLI1 increased sensitivity to inhibitors of both ATR and its key downstream target, CHK1 [160].

In brief, EWS-FLI1 enhances genotoxic stress signaling by disrupting physiologic DNA repair networks.

## 5. Killing the Driver or Taking Advantage of the Driver

### 5.1. Targeting the EWS-FLI1-Related Protein Complex

The treatment of EwS typically consists of radiotherapy, surgical resection, and intensive induction chemotherapy [1,15], which has significantly improved overall survival. However, acute and chronic adverse effects can negatively affect the quality of life of survivors [1].

Knockdown or overexpression of EWS-FLI1 leads to mitotic defects and apoptosis [79,102,112], suggesting that a dynamic fluctuation in EWS-FLI1 expression is essential for EwS cell survival. Therefore, EWS-FLI1 downregulation or overexpression are two potential strategies for EwS.

Targeting EWS-FLI1 directly remains challenging; however, pharmacological targeting of the EWS-FLI1 protein complex, for example using the YK-4-279 and its analog TK-216, is a potentially promising strategy. YK-4-279 blocks the interaction of EWS-FLI1 with helicase A [161,162] or DDX5 [94] and reverses the transcriptional activation of EWS-FLI1. YK-4-279 also leads to an isoform shift from *ARID1A-L* to *ARID1A-S* and prevents the interaction between EWS-FLI1 and *ARID1A-L* [104]. This was originally thought to mimic the inhibition of EWS-FLI1 and showed a synergistic effect with vincristine [163]. However, further work demonstrated that YK-4-279 affects alternative splicing rather than directly inhibiting the transcriptional activity of EWS-FLI1, which mimics the reduction in EWS-FLI1 [94]. Despite the occurrence of drug resistance, as detected in preclinical experiments [164,165], the combination of TK-216 and vincristine showed anti-tumor activity in specific population in a Phase I clinical trial [166] and is well tolerated. A further Phase II clinical trial (NCT02657005) is ongoing. In addition to EWS-FLI1-positive cells, TK216 also exhibits toxicity to other cancer cell lines, such as thyroid cancer [167] and melanoma [168]. Based on the property of cytotoxicity to multiple cancers, Povedano et al. revealed that TK216 binds to distinct sites compared with vincristine and acts as a microtubule-destabilizing agent, such as by providing an additional mechanistic explanation for the clinical activity of this combination [169].

Mithramycin suppresses the transcription of EWS-FLI1 [71]; however, its hepatotoxicity and the narrow therapeutic window results in the discontinuation of a clinical trial (clinical trial: NCT01610570) [170]. Semi-synthetic analogues of mithramycin, such as MTMSA-Trp and MTMSA-Phe, are more selective for EWS-FLI1-positive cell lines [171]. Although mithramycin A does not increase the level of DNA double-strand breaks, it inhibits DNA double-strand break repair and further radiosensitizes EwS [172].

Pretreatment with mithramycin A for 24 h before irradiation significantly reduced clonogenic survival in vitro and delayed tumor regrowth in vivo, prolonging survival of EWS-FLI-positive tumor-bearing mice. An alternative strategy takes advantage of the chimeric protein, leading to R-loops and the positive feedback loop with PARP1 and increasing the sensitivity to PARP inhibitors [24,148]. Mechanistically, EWS-FLI1 blocks the BRCA1, thus mimicking BRCA1 deficiency [24]. However, clinical trials did not meet expectations (clinical trial: NCT01583543) [173]. A clinical trial (NCT01858168) combining olaparib and temozolomide in adults with recurrent/metastatic EwS is ongoing. Chemical genomics screening performed by Iniguez et al. [174] showed that the CDK12/13 inhibitor THZ531 impairs DNA damage repair in a EWS-FLI1-dependent manner, leading to synthetic lethality with PARP inhibitors.

### 5.2. Targeting EWS-FLI1-Related Transcription

In addition to the EWS-FLI1 protein partners, the pharmaceutical targeting of EWS-FLI1-based transcriptome is worth further exploration [33,175]. For example, chemotherapeutic agents targeting LSD1, such as SP-2509, could modify the transcriptional signature of EWS-FLI1 and regulate the proliferation of EwS [31,176]. These agents also show synergic effects with HDAC inhibitors [177,178]. Seclidemstat (SP-2577), a reversible oral LSD1 inhibitor, showed manageable safety and positive preliminary activity in a Phase I trial (NCT03600649) [179], and patients are being recruited for a Phase I/II clinical trial (NCT05266196). EwS is highly sensitive to HDAC inhibition [68,175,180,181,182]. Clinical trials (NCT04308330; NCT00020579) of the VIT (vincristine, irinotecan, and temozolomide) combination have been conducted. However, pan-HDAC inhibitors are highly toxic owing to unspecific effects [183], underscoring the need to develop specific HDAC inhibitors. HDAC1 is critical for the malignancy of EWS-FLI1-mediated transcription and posttranscriptional modifications of EwS [175,184]. Agents targeting Class I/II HDACs show synergistic effects with doxorubicin [106,175], although further clinical research is necessary to avoid potential side effects.

Ecteinascidin 743 (ET-743, trabectedin) interferes with the activity of EWS-FLI1 [185]. However, single-agent trabectedin showed negative results in a clinical trial (NCT00070109) [185,186]. A synergistic effect with SN38 was demonstrated [187], and the combination of trabectedin with irinotecan for the treatment of EwS is being tested in clinical trials (NCT04067115 and NCT02509234). Additionally, a multicohort trial of trabectedin combined with low-dose radiation therapy in advanced/metastatic sarcomas including EwS is also recruiting new patients (NCT05131386).

GSK-J4, an H3K27 demethylase inhibitor, can reverse the H3K27ac driven by EWS-FLI1 at enhancers and shows a synergistic cytotoxic effect when combined with the CDK7 inhibitor THZ1 [188]. BET proteins, particularly BRD4, are also essential for the EWS-FLI1 transcription signature and may be involved in the sensitivity of EwS to JQ1 [32]. The combination of JQ1 with a CDK9 inhibitor, CDKI-73, showed a synergistic effect [34]. The RUVBL1 inhibitor CB6644 also exhibits a synergistic cytotoxic effect with JQ1 in EwS [189].

### 5.3. EWS-FLI1-Dependent Immunotherapy of Ewing Sarcoma

EwS is a ‘cold tumor’ with a low mutation burden, and it is characterized by an immunosuppressive microenvironment [190]. This environment inhibits the infiltration of effective cytotoxic T-cells, which limits the response to clinical treatments. Immunotherapies for EwS mainly consist of immune checkpoint inhibitors and adoptive T-cell transfer [191,192,193,194].

### 5.4. TCR-Based T-Cell Therapy

Peptides derived from the EWS-FLI1 chimeric protein provide unique targets for the TCR-based T-cell therapy, although there is no conclusive evidence supporting their clinical significance [195]. Alternative strategies based on targeting peptides derived from the downstream proteins of EWS-FLI1, such as the CHM1-derived peptide VIMPCSWWV [196,197], have shown clinical regression in EwS patients [193]. The T-cell-targeting LIPI-derived peptides LDYTDAKFV and NLLKHGASL [198], STEAP1-derived YLPGVIAAI [192,194] and MIAVFLPIV [199], PAPPA-derived IILPMNVTV [200], EZH2-derived YMCSFLFNL [197], PAX3-derived QLMAFNHLI, and the modified version, QLMAFNHLV [201], showed effective killing of HLA-A*02:01^+^ ES cell lines. XAGE-1A is a direct downstream target of EWS-FLI1 (Appendix A) and serves as a potential target in TCR-based T-cell therapy [202,203], although no killing of targeting EwS has been identified yet [204].

## 6. CAR-T Therapy

Unlike TCR-based T-cell therapy, which is limited to specific HLA restriction and deficient HLA expression in EwS [205] because of the presence of myeloid-derived suppressor cells, F2 fibrocytes, and M2-like macrophages in the microenvironment [190], CAR-engineered T-cell therapy can target specific cell-surface antigens in tumors, independent of HLA. VEGFR2 is a potential target for CAR-T-cell therapy in EwS [206]. In addition to TCR-T-cells, CAR-T-cells [207] can target STEAP1, which is involved in the malignant phenotype of EwS [65].

CAR-T targeting GPR64, ROR1, and IGF1R, which are highly expressed in EwS [137,208], leads to a selective killing of EwS in vivo [209]. LINGO1, which is highly expressed in EwS [210], is a direct target of EWS-FLI1 (Appendix A). EZH2 inhibition by GSK-126 induces GD2 surface expression in EwS [211], and the combination of CAR-T therapy targeting GD2 and EZH2 inhibitors have synthetic cytotoxic in the treatment of EwS; this kind of combination provides new options for the clinical application. IL1RAP, a direct target of EWS-FLI1, is highly expressed in EwS, but minimally expressed in normal tissues, which makes it a promising surface target for EwS [70] and a potential candidate for advanced CAR-T therapy. ICAM-1 can promote tumor cell/T-cell interaction and T-cell activation, and the knockdown of EWS-FLI1 upregulates ICAM-1 expression and leads to the upregulation of PD-L1 and PD-L2, both proteins that inhibit the activity of T-cells [115]. Blocking PD-1 with a checkpoint inhibitor could increase the T-cell-mediated killing of EwS cells with low expression of EWS-FLI1.

## 7. Perspectives

Strategies targeting EWS-FLI1 for the treatment of EwS need to be established. Direct targeting of EWS-FLI1 or therapies related to EWS-FLI1 protein are both viable options. In the ‘EWS-FLI1 toxicity’ strategy, targeting EWS-FLI1 offers a wide therapeutic window because of its unique expression pattern. However, further work is necessary to develop specific inhibitors. Additionally, a better understanding of the cytotoxicity and chemoresistance of YK-4-279, such as its destabilization in microtubules, is necessary to optimize its clinical application. EWS-FLI1 contributes to the sensitivity of EwS to radiotherapy and PARP inhibitors and is involved in DNA damage. Therefore, further research on the EWS-FLI1 protein complex is necessary to elucidate its role in chemoresistance, particularly the dynamic balance of EWS-FLI1 with wild-type EWSR1 and their related phase transitions.

Targeting the downstream factors of EWS-FLI1 with TCR- or CAR-T-based immunotherapy is based on the role of EWS-FLI1 in modulating CHM1, IL1RAP, and other molecules (Figure 3A). This type of immunotherapy has revolutionized cancer treatment, whereas canonical methods of retro- or lentivirally induced TCR may disrupt TCR dynamics and lead to atypical T-cell function.

Recent work from Schober et al. demonstrated that orthotopic TCR replacement using nonviral methods results in near-physiological T-cell function [212], thereby preventing potential mispairing of endogenous and extraneous TCR. Our work also provides evidence that CRISPR/Cas9-engineered TCR replacement by TCR targeting CHM1 in T-cells could enhance the longevity of the cytotoxic effect of targeting EwS compared with the transgenic T-cells via retrovirus [213]. Metabolic or epigenetic reprogramming to enhance the function of T-cells provides another therapeutic option [214]. Apart from leading apoptosis, the PARP inhibitor modulates the tumor microenvironment by activating the cGAS-STING pathway and, thus, enables low-dose CAR-T-cells with higher efficacy [215,216]. Olaparib also suppresses myeloid-derived suppressor cell migration via the SDF1α/CXCR4 axis and promotes the survival of CD8+ T-cells in tumor tissue [217]. The combination of PARP inhibitor and CAR-T/TCR-T therapy is worthy of further research in EwS.

Through multiple pathways, the purpose of T-cell therapy is to achieve the apoptosis of cancer cells [216]. LSD1 inhibition promotes tumor immunogenicity and T-cell infiltration [218]. Genes negatively correlated with LSD1 in EwS are mainly associated with immune responses (Figure 3B). LSD1 expression in EwS is the highest among all cancers, and LSD1 inhibition leads to robust apoptosis through the endoplasmic reticulum (ER) stress pathway in EwS [176]. Exploring the combination of LSD1 inhibition with immunotherapy targeting the downstream of EWS-FLI1 could lead to important discoveries.

## 8. Conclusions

EWS-FLI1 was identified three decades ago (1992), 70 years after James Ewing first defined Ewing sarcoma (1921). Since then, this chimeric protein has been widely accepted as the oncogenic driver and master regulator of EwS because of its ability to modify the transcriptome. However, pharmacologically targeting EWS-FLI1 remains challenging because of its disordered structure. The clinical treatment of EwS mainly consists of radiotherapy, surgical resection, and intensive induction cytotoxic chemotherapy, which can lead to acute and chronic adverse effects.

Two strategies have been proposed based on the concept of precision medicine: targeting EWS-FLI1 directly or developing methods that indirectly exploit this dominant oncogene. This can involve targeting EWS-FLI1 in DNA damage or immunotherapy targeting the downstream transcripts such as CHM1. In this review, we summarized the current knowledge of EWS-FLI1 and the advances in related therapies, thereby providing the basis for further research.

## Figures and Tables

**Figure 1 cancers-15-04035-f001:**
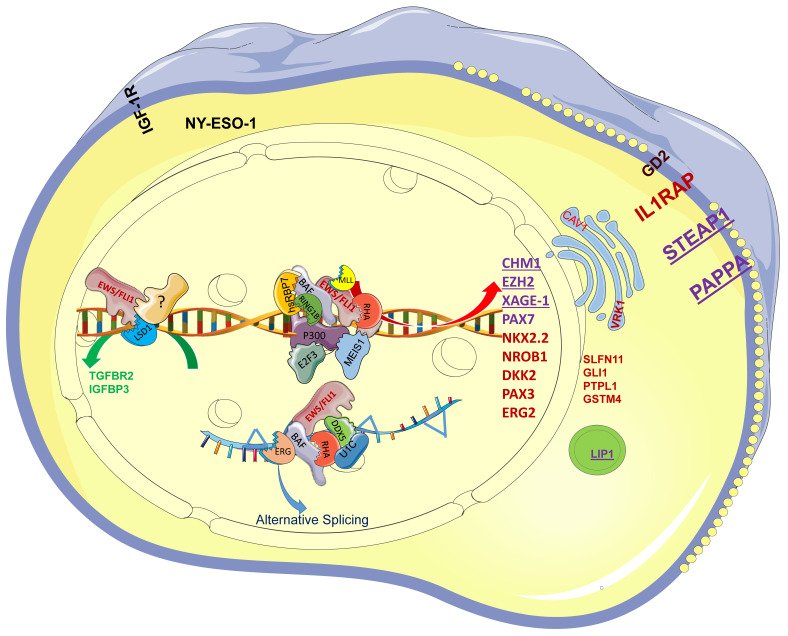
The EWS-FLI1 protein complex drives the specific transcription profile of EwS. EWS-FLI1 recruits E2F3, hsRBP7, BAF, RING1B, RHA, P300, and MEIS1, among others, to GGAA repeats and further activates CHM1, EZH2, PAX7, NKX2.2, NROB1, and STEAP1, among others. EWS-FLI1 functions as a protein complex with ERG, BAF, RHA, DDX5, and U1C to drive alternative splicing. Among the genes, the purple ones, such as CHM1, could serve as TCR-based immunotherapy targets; the red ones, such as NKX2-2, serve as diagnostic markers in clinic diagnosis. EWS/FLI1 recruits LSD1 and unknown transcription factors (?) to repress TGFBR1 and IGFBP3, which still needs further research.

**Figure 2 cancers-15-04035-f002:**
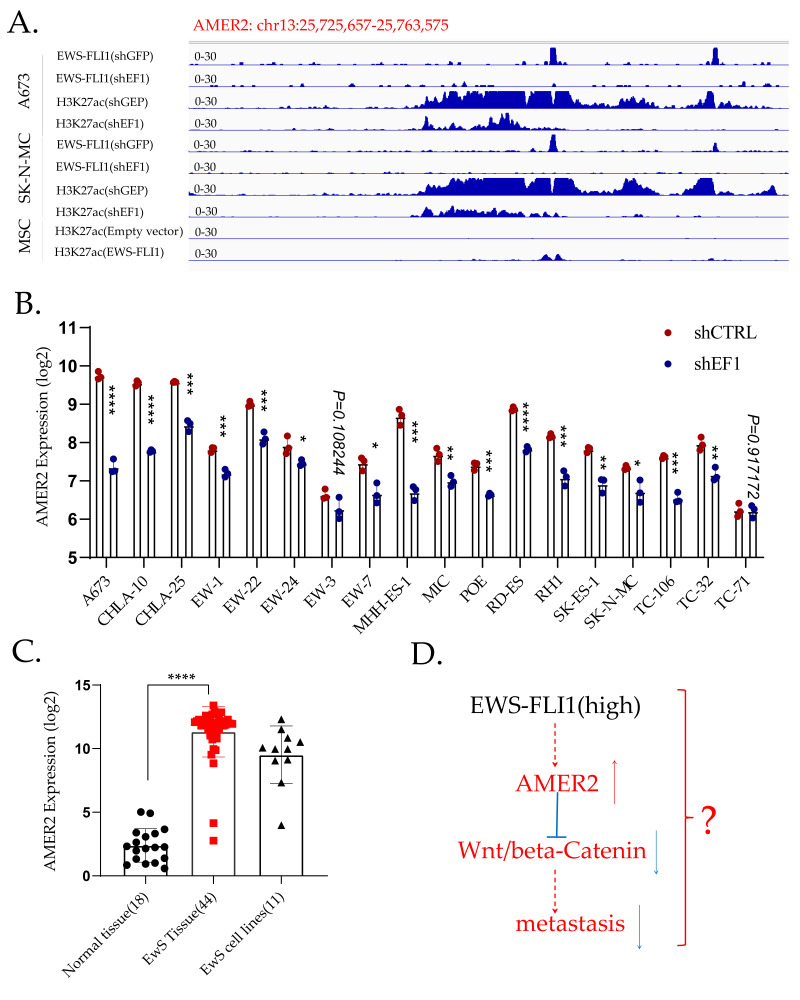
AMER2 is a direct target of EWS-FLI1. (**A**). CHIP-sequencing data demonstrate that AMER2 is a direct target of EWS-FLI1. In A673 and SK-N-MC cells, EWS-FLI1 binds to the promotor of AMER2 directly, and the H3K27ac level is lower after EWS-FLI1 knockdown. In MSC, transduction of EWS-FLI1 enhances the H3K27ac level in the promotor region of AMER2. (**B**). Knockdown of EWS-FLI1 downregulates AMER2 in EwS cell lines apart from the TC-71 cell line. (**C**). AMER2 expression is higher in EwS tissues than in normal tissues. (**D**). Potential mechanism of AMER2 in the metastasis of EwS. *, *p* < 0.01; **, *p* < 0.05; ***, *p* < 0.001; ****, *p* < 0.0001.

**Figure 3 cancers-15-04035-f003:**
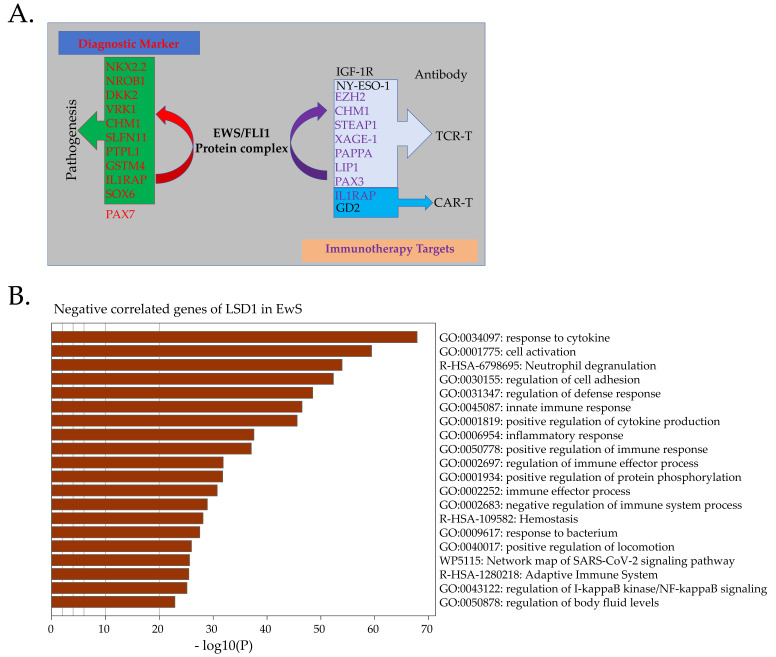
Immuno-targeting therapy for EwS. (**A**). Direct downstream targets of EWS-FLI1, such as CHM1, STEAP1, and PAPPA, could serve as TCR-based immunotherapy. IL1RAP and STEPA1 are potential targets for CAR-T therapy. NY-ESO-1 is not a direct target of EWS-FLI1, but it is also a good target for TCR-T therapy. NKX2-2, NROB1, and PAX7 are reliable diagnostic markers of EwS. (**B**). Negatively correlated genes of LSD1 (*p* < 0.05) in EwS samples are mainly enriched in immune responses.

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
