# Peer review of "Targeted Therapy for EWS-FLI1 in Ewing Sarcoma"

_cancers, 2023, doi:10.3390/cancers15164035_

Round 1
Reviewer 1 Report
This manuscript summarized roles of EWS-FLI1 in Ewing sarcoma (EwS), therapies which target EWS-FLI1 or its related proteins, and the CAR-T therapy that are being developed to treat EwS. Overall, it included a relative comprehensive discoveries which have been reported. The manuscript is relatively well-structured. The language needs to be improved. My comments are following.
1. A review article published by Li & Chen discussed mechanism of EWS-FLI1 in reprograming epigenome and inducing transcriptional changes (https://pubmed.ncbi.nlm.nih.gov/35740349/). Since it is the most recent review which discussed mechanism of EWS-FLI1 in mediating EwS, this review should be cited.
2. Line 30, the cellular origin of EwS is derived from neuroectodemal cells or primitive “mesenchymal stem cells” instead of “mesenchymal cells”.
3. In line 81-85, the genes which are activated by EWS-FLI1 are listed. However, the authors missed MYC. MYC is also a downstream target of EWS-FLI1 (https://pubmed.ncbi.nlm.nih.gov/8164678/). Recently, Li et al also reported therapies targeting MYC and its interacting protein RUVBL1 (https://pubmed.ncbi.nlm.nih.gov/37075745/) to treat EwS. Considering importance of MYC in regulating tumors including EwS, the authors should include MYC and at least the two publications listed here. For example, using of CB6644 and JQ1 to treat EwS reported by Li et al (https://pubmed.ncbi.nlm.nih.gov/37075745/) can be cited at line 277. MYC is a downstream of EWS-FLI1 can be cited at line 85.
4. In “2.4 EWS-FLI1 in the metastasis in EwS” part, the manuscript also has missing citations. For example, a review article titled “Regulation of Metastasis in Ewing Sarcoma” (https://pubmed.ncbi.nlm.nih.gov/36230825/) discussed how EWS-FLI1 contributes to the heterogeneity and metastasis of EwS. Therefore, it should be cited.
5. Line 154, “cytosine arabinoside (ARA-C) leads been shown to downregulate...” does not make sense. Do the authors want to say “cytosine arabinoside (ARA-C) has been shown to downregulate...”?
6. Line 175, “enriched in a HIF-1α signature” does not make sense. I don’t understand what the sentence is saying.
7. Line 178-185 paragraph, it needs to be re-written.
8. Line 194, “the hypersensitive to radiation and DNA damaging ...” does not make sense. Hypersensitive is adjective but not a noun. Therefore, hypersensitive can’t provides further evidence...
9. Line 200, EWS-FLI1 maintains the expression of what? Do authors want to say “EWS-FLI1 maintains the expression of PARP-1”?
10. Line 198 - 215, it would be great if authors can use one or two sentences to conclude this paragraph.
11. Ling 345, “70 years after 70 years after” is redundant.
Language needs to be further edited extensively.
Author Response
Dear Reviewer,
We profoundly appreciate esteemed your efforts on our manuscript. We completed all requested information and addressed all comments as detailed below in blue. All changes in the manuscript Word doc are given in track mode. Please accept our gratitude to the editor and reviewers for their competent advice. Please see the attachment file, which includes our feedback and English editing certificate. Thanks very much!

Reviewer 2 Report
The review manuscript titled EWS-FLI1 in Ewing Sarcoma and Targeted Therapy b y Gong et al., is an interesting topic where authors discuss various aspects of EWS-FL1 in a rare cancer, namely Ewing Sarcoma. Overall, the manuscript is nicely written, contains pertinent and important information. The authors have used adequate number of related and important articles to support their statements. The only concern is lack of discussion on the gene products regulated by EWS-FL1, specially those involved in apoptosis and what role they play in sensitivity/resistance to CAR T cell therapy
moderate English editing needed
Author Response
Dear Reviewer,
We profoundly appreciate esteemed your efforts on our manuscript. We completed all requested information and addressed all comments as detailed below in blue. All changes in the manuscript Word doc are given in track mode. Please accept our gratitude to the editor and reviewers for their competent advice. Please see the attachment which is our feedback. Thanks very much. We also performed English editing with a certificate

Reviewer 3 Report
This is an interesting paper about EWS/Fli1 translocation in Ewing-sarcoma, its possible role in intracellular mechanisms leading to tumor phenotype and as a possible target of therapy. This is a real hot topic in Ewing-sarcoma as there is no real new effective therapeutic target in this disease. Authors give a good overview to understand in which pathways as possible therapeutic approaches could be involved the translocation.
Minor comments:
the English of the paper is sometimes disturbing and impair understanding
(e.g. line 16: "a rare highly pediatric malignancy"
line 27: "a poorly differentiated malignant that mainly.."
line 194: "The hypersensitive to radiation..."
Form of References is generally incorrect, which should be corrected
the English of the paper is sometimes disturbing and impair understanding
(e.g. line 16: "a rare highly pediatric malignancy"
line 27: "a poorly differentiated malignant that mainly.."
line 194: "The hypersensitive to radiation..."
Author Response
Dear Reviewer,
We profoundly appreciate esteemed your efforts on our manuscript. We completed all requested information and addressed all comments as detailed below in blue. All changes in the manuscript Word doc are given in track mode. Please accept our gratitude to the editor and reviewers for their competent advice. Please see the attachment including our feedback and English editing certificate.

Reviewer 4 Report
I would not recommend publishing this review due to the following three main reasons: (1) Many sentences throughout this manuscript are generally hard to interpret. There are many mistakes, unconventional writings and grammar errors throughout the manuscript. (2) This review misses many key recent findings that provide therapeutic implications. For example, a most recent paper (Lu et al 2023, PMID: 36658220) describes that ETV6 constrain EF to promote EwS metastasis. In addition, cohesin components, i.e. STAG2, play a big role in EWS-FLI1-driven EwS (PMID: 34129824, 33930311). Many dependencies have not been described in this review manuscript. (3) This review fails to provide new clinical inspiration with mechanistic aspects. Although the authors list many ongoing clinical trials, some similar review articles have already done so, and there lacks inspiring discussions and opinions about those clinical studies and trials, which connects the former part of the mechanism of targeting EF. The MOA of YK-4-279 is still unclear however there is in vivo toxicity indicated in studies. Again, many citations are missing for the MOA of the drugs listed. For example, since the authors mentioned the trial of TK216 combined with vincristine, they failed to cite Povedeno et al 2022 (PMC9394687), which underlies the mechanism of treatment.
Too many aspects need to be scrutinized thereby unfortunately it is not recommended to have this manuscript published at the current format.
Review comments
In the review EWS-FLI1 in Ewing Sarcoma and Targeted Therapy, First, Gong et al provided a general overview on causes and consequences at the molecular level for Ewing Sarcoma. Then, they reviewed the role of EWS-FLI1 in transcription regulation. Lastly, they provided a brief summary of the current treatments for EWS-FLI1 and gave future perspectives on precision medicine for more effective treatments. The review manuscript covers a broad range of topics in the EWS-FLI1 field, however, many of the points made in the paper lacks coherent context and more in-depth explanation. In addition, the syntax of the language is poorly written. Thus, I will not be supportive for publishing this review paper.
There are several comments listed below:
Section 2 should have a more accurate heading because beside transcription, epigenetic reprograming and alternative splicing, it also included information such as protein turnover, metastasis, and effect on DNA damage. These topics should be discussed separately outside of section 2 heading.
Figure 1 and related text: It is rare that people use et al to depict a figure. A review should not be simply listing relevant literatures. Many of the EWS-FLI1 interacting proteins listed in this section had little context or background information and thus made readers challenging to coherently understand. The author could simplify figure 1 into a more generic illustration or expand on a couple of specific examples mentioned in this section. Many elements in the figure are not explained in the legend, for example, what does the question mark stand for? What is the difference between genes highlighted in purple vs. red?
Section 2.3 title might be misleading. Turnover often refers to protein synthesis and degradation, but in this section transcriptional regulation of EWS-FLI1 is mostly emphasized.
Figure 2A serves little to its purpose and could be taken out. Figure 2B was not explained in details in the legend. What are those cell lines, what are the different conditions in each cell line, what antibodies were probed? Also, the scale for each lane is different. Figure 2E, what are the differences between faint line arrows and dash line arrows? What does the question mark mean? None of these were explained.
Section 2.5, many of the recent publications that are important to the role of EWS-FLI1 in causing DNA damage are not mentioned or cited.
In section 3, many of the drugs were mentioned for EWS-FLI1 treatment, again very little details were given on the mechanism of action for the drug.
Figure 3B, what data or samples were used for the GO term analysis? How was the analysis done? There should be more details explaining the figure.
The authors need to check typos and inaccurate wordings throughout text to make comprehensive statements. For example, “rare highly pediatric malignancy”; “quiet genome background”, “manipulates multiple proteins” etc. Also, there are many extra spaces/gaps between sentences that need to be reformatted.
The English needs to be significantly improved to make the points clear. The flow of the paper needs to be improved as well. I’d suggest that the language should be checked by professional writers or institutes before being published.
Author Response

(The authors gave the same response as above.)

Round 2
Reviewer 4 Report
After the revision, the manuscript has been significantly improved and now it is more readable and scientifically sound. There are several recommendations and minor edits that are still required for the final version.
Line 118: "By contrast" should be "In contrast"
The big comment that I have for this paper is the title. Probably "EWS-FLI1 in Ewing Sarcoma Targeted Therapy" or "Targeted Therapy for EWS-FLI1 in Ewing Sarcoma" would be a better option (remove the "and"). The current title is a bit mis-leading because as a scientist in the field, one could expect to read a thorough review for EF biology and at the same time how to target it for therapy for ES. However, as a review, it is really neither a thorough review for EWS-FLI1 in Ewing sarcoma nor a review for targeted therapy. But it does angle towards reviewing targeting EF in Ewing sarcoma, which in that sense is a valuable review. By changing the title, it could easily fix the problem.
Citations are needed in Lines 210, 303.
Line 263-264: EWS-FLI1 causes DNA damage and growth arrest. Su et al 2021 (PMID: 33766983) also show that it causes replication stress and growth arrest in many primary cell lines. This paper should also be cited in line with citations 113, 144, 145, on line 264.
Line 317: remove "a" before "challenging.
The English has been significantly improved.
Author Response
Dear Review,
We are very grateful for your professional comments on our manuscripts! We completed the requested information and addressed all comments as detailed below in yellow. All changes in the manuscript word. doc are given in track mode.
Please see the attachment file, which is the cover letter. Thanks very much!
